# SWITCHED LINEAR PROJECTIONS AND INACTIVE STATE SENSITIVITY FOR DEEP NEURAL NETWORK INTERPRETABILITY

## ABSTRACT

We introduce *switched linear projections* for expressing the activity of a neuron in a ReLU-based deep neural network in terms of a single linear projection in the input space. The method works by isolating the active subnetwork, a series of linear transformations, that completely determine the entire computation of the deep network for a given input instance. We also propose that for interpretability it is instructive and meaningful to focus on the patterns that deactive the neurons in the network, which are ignored by the exisiting methods that implicitly track only the active aspect of the network's computation. We introduce a novel interpretability method for the *inactive state sensitivity* (Insens). Comparison against existing methods shows that Insens is robust (in the presence of noise), complete (in terms of patterns that affect the computation) and a very effective interpretability method for deep neural networks.

## 1 INTRODUCTION

It is notoriously hard to interpret how deep networks accomplish the tasks for which they are trained. At the same time, due to the pervasiveness of deep learning in numerous aspects of computing, it is increasingly important to gain understanding of how they work. There are risks associated with the possibility that a neural network might not be "looking" at the "right" patterns (Nguyen et al.; Geirhos et al., 2019), as well as opportunities to learn from the network's capable of *better than human* performance (Sadler & Regan, 2019). Hence, there is ongoing effort to improve the interpretation and interpretability of the internal representation of neural networks.

What makes this interpretation of the inside of a neural network hard is the high dimensionality and the distributed nature of its internal computation. Aside from the first hidden layer, neurons operate in an abstract high-dimensional space. If that was not hard enough, the analysis of individual components of the network (such as activity of individual neurons) is rarely instructive, since it is the intricate relationships and interplay of those components that contain the "secret sauce". The two broad approaches to dealing with this complexity is to either use simpler interpretable models to approximate what a neural network does, or to trace back the elements of the computation into the input space in order to make the internal dynamics relatable to the input. In the latter approach we are typically interested in neurons' *sensitivity* – how the changes in network input affect their output, and *decomposition* – how different components of the input contribute to the output.

In this paper we propose a straightforward and elegant method for expressing the computation of an arbitrary neuron's activity to a *single linear projection* in the input space. This projection consists of a *switched weight vector* and a *switched bias* that easily lend themselves to sensitivity analysis (analogous to gradient-based sensitivity) and decomposition of the internal computation. We also introduce a new approach for interpretability analysis, called *inactive state sensitivity* (Insens), which uses switched linear projections to aggregate the contribution of patterns in the input that *deactivate* neurons in the network. We demonstrate on several networks and image-based datasets that Insens provides a comprehensive picture of a deep network's internal computation. The only constraint for the proposed methods is that the network must use ReLU activation functions for its hidden neurons.

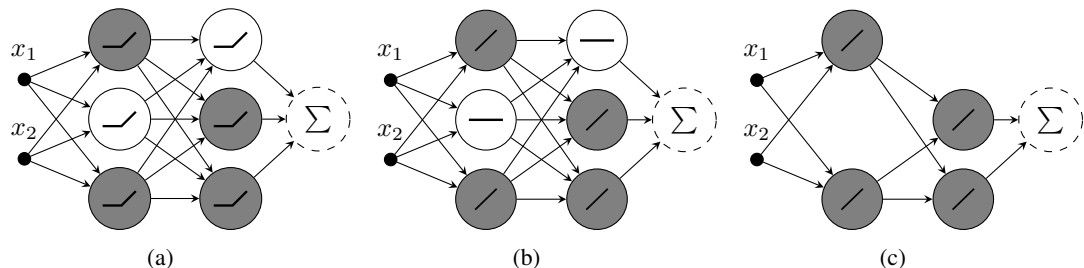

Figure 1: Let's assume that for a particular input $[x_1\ x_2]$ going into the ReLU network shown in (a) the white neurons are *inactive*; then, for this particular input, the network from (a) is equivalent to network in (b) where the inactive neurons are treated as *dead* and the *active* ones operate in the linear part of their ReLU activation fuction; which makes both of these networks equivalent to one in (c); the grey hidden neurons form the active subnetwork.

## 2 RELATED WORK

Previous work on deep learning interpretability is extensive with a wide variety of methods and approaches – Simonyan et al. (2014); Zeiler & Fergus; Bach et al. (2015); Mahendran & Vedaldi; Montavon et al. (2017); Sundararajan et al. (2017); Zhou et al. (2019) being just a selection of the most prominent efforts in this area. Our work on the single linear projection follows the approach akin to Lee et al. and Erhan et al. (2009), where the objective was to interpret the computation performed by an arbitrary neuron for a particular input vector as a projection in the input space. However, whereas these previous attempts were based on Deep Belief Nets (Hinton et al., 2006) and required an approximation of the said projection, our method is a forward computation that gives the neuron's activity in terms of a linear projection in the input space. It works for any neural network, including convolutional ones, as long as all hidden neurons use piecewise linear activation functions.

All existing methods for interpretability of deep learning, due to the nature of ReLU computation, necessarily provide information only about the active subnetwork of the ReLU-based architecture. Our own observations, as well as other evidence showing that in practice neural networks produce a relatively low number of activation regions (Hanin & Rolnick, 2019), lead us to the hypothesis that the analysis of the patterns in the input that *switch neurons off* gives an excellent picture of a network's sensitivity. We also take the view that too much *interpretation* in interpretability introduces the risk of showing us what we expect to see and *not* what the network is actually focussing on. For instance, in Deep Taylor Decomposition (Montavon et al., 2017) choices of different root-points for the decomposition of the relevance function lead to different rules for Layerwise Relevance Propagation (LRP)(Bach et al., 2015), which can lead to different interpretations of what is important in the input. The LRP-$\alpha_1\beta_0$ rule, for example, emphasises the computation over the positive weights in the network while discounting the relevance of the information passing through the negative weights. This rule is justified by assumptions about desired properties of the explanation, but this comes with a risk of confirmation bias. Insens is an attempt to take into account the patterns in the input that *cause* the neurons inside the network to produce zero output. The information related to the inactive network may seem irrelevant, since inactive neurons do not directly contribute to the computation of the overall output. However, there is *something* in the input that switches a particular set of neurons off, thus regulating the active computation, and as we show in this paper, this *something* carries a lot of meaningful information.

## 3 SWITCHED LINEAR PROJECTIONS

The basis of the switched network concept is the fact that neurons that produce output of zero do not contribute to the computation of the overall output of the network. The notion of *dead* neurons, that is neurons that always output zero, is not new, nor is the realisation that these neurons, along with their connecting weights, can be taken out the network without any impact on the computation. In a switched projection, we treat the zero-output neurons as temporarily *dead* for a given instance of input. We refer to these neurons as *inactive*, since they may become *active* for a different network input. Thus we isolate the subnetwork of the active neurons in a given computation. As it happens,

for a ReLU neuron the active neurons are those that pass their activity, the weighted sum of its inputs plus bias, directly to its output[1]. This means that a subnetwork of active ReLU neurons is just a series of linear transformations, which is equivalent to a single linear transformation. As a result, we can express the computation performed by any neuron in a ReLU network as a projection onto a switched weight vector in the input space plus the switched bias. The term *switched* indicates that this weight and bias vector changes when the state of the network changes, the state corresponding to the particular combination of the active and inactive neurons in the network. Figure 1 illustrates the concept graphically, and a formal description is given in the following proposition:

**Proposition 1** (Switched linear projections). *Let* $\mathbf{x} \in \mathcal{R}^d$ *be a vector of inputs,* $\mathbf{w}_{li} \in \mathcal{R}^{U_{l-1}}$ *the weight vector, and* $b_{li} \in \mathcal{R}$ *the bias of neuron* $i$ *in layer* $l$ *(with* $U_{l-1}$ *inputs from the previous layer). Let the activity of a neuron* $i$ *in layer* $l$ *be defined as:*

$$v_{li}(\mathbf{x}) = \Big( \ldots \sigma_r\big(\sigma_r(\mathbf{x}\mathbf{W}_1 + \mathbf{b}_1)\mathbf{W}_2 + \mathbf{b}_2\big) \ldots \Big)\mathbf{w}_{li} + b_{li}, \qquad (1)$$

*where* $\mathbf{W}_l = \begin{bmatrix} \mathbf{w}_{l1}^T & \ldots & \mathbf{w}_{lU_l}^T \end{bmatrix}$, $T$ *denotes transpose,* $\mathbf{b}_l = \begin{bmatrix} b_{l1} & \ldots & b_{lU_l} \end{bmatrix}$ *and* $\sigma_r(v) = \max(v, 0)$ *is the ReLU activation function. If we define an input-dependent state of the network as*
$\mathbf{W}_l^{(\mathbf{x})} = \begin{bmatrix} \dot{\sigma}_r\big(v_{l1}(\mathbf{x})\big)\mathbf{w}_{l1}^T & \ldots & \dot{\sigma}_r\big(v_{lU_l}(\mathbf{x})\big)\mathbf{w}_{lU_l}^T \end{bmatrix}$ *and*
$b_l^{(\mathbf{x})} = \begin{bmatrix} \dot{\sigma}_r(v_{l1}(\mathbf{x}))b_{l1} & \ldots & \dot{\sigma}_r(v_{lU_l}(\mathbf{x}))b_{lU_l} \end{bmatrix}$,
*where* $\dot{\sigma}_r(v) = \frac{d\sigma_r(v)}{dv}$, *then for*

$\widehat{\mathbf{w}}_{li}^T(\mathbf{x}) = \mathbf{W}_1^{(\mathbf{x})}\mathbf{W}_2^{(\mathbf{x})} \ldots \mathbf{W}_{l-1}^{(\mathbf{x})}\mathbf{w}_{li}^T$ *and*

$\widehat{b}_{li}(\mathbf{x}) = \mathbf{b}_1^{(\mathbf{x})}\mathbf{W}_2^{(\mathbf{x})} \ldots \mathbf{W}_{l-1}^{(\mathbf{x})}\mathbf{w}_{li}^T + \mathbf{b}_2^{(\mathbf{x})}\mathbf{W}_3^{(\mathbf{x})} \ldots \mathbf{W}_{l-1}^{(\mathbf{x})}\mathbf{w}_{li}^T + \ldots + \mathbf{b}_{l-1}\mathbf{w}_{li}^T + b_{li}$, *we have*

$$v_{li}(\mathbf{x}) = \mathbf{x}\widehat{\mathbf{w}}_{li}^T(\mathbf{x}) + \widehat{b}_{li}(\mathbf{x}). \qquad (2)$$

The proof is provided in Appendix A. Note that the ReLU derivative, $\dot{\sigma}_r(v)$, is just a convenient definition for a step function, so that

$$\dot{\sigma}_r(v)\mathbf{w} = \begin{cases} \mathbf{w} & , v > 0 \\ \mathbf{0} & , \text{otherwise}. \end{cases} \qquad (3)$$

To simplify the notation, whenever referring to the parameters of the switched projection $\widehat{\mathbf{w}}, \widehat{b}$, as well as activity $v$, we will drop the explicit dependency on $\mathbf{x}$.

While Figure 1 illustrates the switching concepts on a small fully connected network, switched linear projections can be computed for networks with convolutional as well as pooling layers. A convolutional layer is just a special case of a fully connected layer with many weights being zero and groups of neurons constrained to share the weight values on their connections. For max pooling, the neurons that do not win the competition, and thus their output does not affect the computation from then on, are deemed to be *inactive* regardless of the output they produce.

### 3.1 SENSITIVITY

Equation 2 makes it obvious that a given neuron's switched weight vector is just the derivative of its activity with respect to the network input, $\widehat{\mathbf{w}} = \frac{\partial v(\mathbf{x})}{\mathbf{x}}$. Thus, the switched weight vector is analogous to gradient-based sensitivity analysis. Figure 2 shows the heatmaps of the switched weight sensitivity for the same set of hidden neurons with different inputs from the MNIST-trained 2CONV neural network (for details on the network architecture featured in this paper see Appendix B). In this visualisation we show normalised $\widehat{\mathbf{w}}$, with intensity of red corresponding to the larger positive value, and the intensity of blue the negative value. The neurons were chosen from the 2nd convolutional layer and the penultimate fully connected layer respectively such that for the four

---

[1]In our terminology, activity denotes output before the activation function and an active neuron is one that produces non-zero output after the activation function; for a ReLU neuron the active and inactive neurons are those that have positive and negative activity respectively.

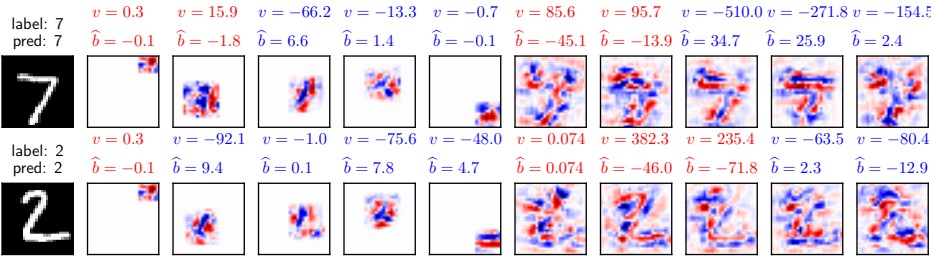

Figure 2: The heatmaps of $\widehat{\mathbf{w}}$ of arbitrary neurons from 2 different MNIST inputs (first column), in the second convolutional layer (next 5 columns) and the first fully connected layer (last 5 columns) of the 2CONV neural network; normalised intensity of red and blue in each heatmap corresponds to the magnitude of the positive and negative value (with white indicating 0); the activity $v = \mathbf{x}\widehat{\mathbf{w}}^T + \widehat{b}$ and the switched bias $\widehat{b}$ are shown above the heatmap of each neuron.

considered inputs some neurons were always active, some inactive, and others sometimes active and sometimes not.

Figure 2 makes it clear that a given neuron is not necessarily sensitive to the same pattern for different network inputs – this is most evident in the sensitivity of the neurons of the fully connected layer. Also note that some neurons in the convolutional layers are active despite the fact that they only "see" the part of the input image that is "empty" (all pixels are black). This leads us to the conclusion that a given neuron is not necessarily a detector of a particular pattern in the input space, which is often the underlying assumption of the existing interpretability techniques.

Something also apparent in our switched projection analysis, though not evident from Figure 2, is that for a given input most of the neurons in the network are inactive. On average, only 17% of the neurons were active for a given MNIST input in this architecture. The fact that only a subset of neurons are active in a given computation is not a quirk of one specific network, as observed by (Hanin & Rolnick, 2019). Switched linear projections give us an interpretation of a deep network as a set of linear, input-dependent, transformations. Something about the input activates a subset of neurons, but also keeps all the remaining neurons inactive. Traditional sensitivity analysis, as well as the one shown in Figure 2, shows a direction (or magnitude) of the gradient of the input that would increase neurons activity provided the same state of the network remained unchanged. However, often in these networks the state does change even after small perturbations of the input, often resulting in same classification, but different input gradient. We reason that the analysis of the state, including information about what makes the neurons inactive, provides a missing and likely very important aspect, to interpretability than analysis of the active subnetwork alone.

## 3.2 DECOMPOSITION

Switched linear projections can decompose the activity of a neuron into contributions from its input, in our case the pixels. For this we propose another re-interpretation of the computation of the output that will allow us to distribute the bias over the attributes of the input vector. Note that for a linear projection

$$v = \mathbf{x}\mathbf{w}^T + b = (\mathbf{x} - \mathbf{c})\mathbf{w}^T, \tag{4}$$

where $\mathbf{c} = \mathbf{x} - \frac{v}{\mathbf{w}\mathbf{w}^T}\mathbf{w}$, $b = -\mathbf{c}\mathbf{w}^T$, and $\mathbf{c} \in \mathcal{R}^d$. The vector $\mathbf{c}$ can be thought of as a translation of the coordinate system to a neuron-centered one, where $\mathbf{w}$ goes through the origin at $\mathbf{c}$. Montavon et al. (2017) call this vector *the nearest root point*, but we will refer to it as the neuron's *centre*. Figure 3 provides a geometric interpretation of $\mathbf{c}$ in a simple 2D scenario. Since $\mathbf{c} \in \mathbf{R}^d$, we can break down the computation of $v$ such that

$$v = \sum_{j=1}^{d}(x_j - c_j)w_j, \tag{5}$$

and take $(x_j - c_j)w_j$ to be the contribution of the input component $j$ (in our examples, a single pixel) to the computation of neuron's activity. Since the switched linear projection is equivalent to

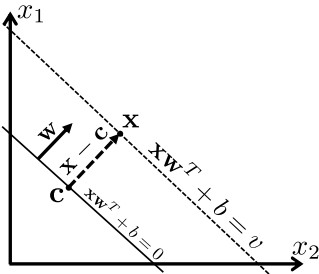

Figure 3: Graphical representation of the concept of a neuron's centre $\mathbf{c}$ in a 2D scenario, where input vector is $\mathbf{x} = [x_1\ x_2]$; the diagram shows input-space coordinate system – the neuron-specific one would have its origin at $\mathbf{c}$.

weights and bias of a single neuron, we can compute the *switched centre*

$$\widehat{\mathbf{c}} = \mathbf{x} - \frac{v}{\widehat{\mathbf{w}}^T \widehat{\mathbf{w}}} \widehat{\mathbf{w}}. \tag{6}$$

The switched centre $\widehat{\mathbf{c}}$ is related to the concept of *reference* in DeepLIFT (Shrikumar et al., 2017) that gets subtracted from the input in order to extract a pattern of interest. However, whereas in DeepLIFT the *reference* is external to the model, and used for examination of perturbations it induces in the output, our proposed centre is a component of the actual computation of the network's output; one could say, $\widehat{\mathbf{c}}$ is a given neuron's inherent *reference*.

Visualisations based on decomposition of active neurons do not offer anything more than existing methods. However, the concept of neuron's centre will be used in the method we propose next, for interpretability based on the sensitivity of the inactive network.

## 4 INACTIVE STATE SENSITIVITY

Since a switched linear projection can be found for any neuron in the network, it can just as easily relate what it is about the input that drives a neuron into the negative just as well as positive. Tracking the patterns in the input that make the neuron more inactive tells us about the aspects of the input that would ensure the stability of the network's state. The bigger the magnitude of the negative activity, the less likely the inactive neurons are to switch on, and thus change the switched projection. Some of the inactive neurons are closer and others further away from the point where they would activate. Hence, we propose a definition of inactive sensitivity based on switched linear projection that takes the magnitude of activity into account,

$$\widehat{\boldsymbol{\omega}}_i = \mathbf{x} - \widehat{\mathbf{c}}_i = \frac{v_i}{\widehat{\mathbf{w}}_i \widehat{\mathbf{w}}_i^T} \widehat{\mathbf{w}}_i. \tag{7}$$

The difference $\mathbf{x} - \widehat{\mathbf{c}}_i$ can be thought of as the component of the input that projects onto a neuron's switched weight vector $\widehat{\mathbf{w}}_i$ in the coordinate system centred at $\widehat{\mathbf{c}}_i$. Note that $\widehat{\boldsymbol{\omega}}_i$ is still a vector in the direction of $\widehat{\mathbf{w}}_i$, and so it is a measure of a neuron's sensitivity, but its magnitude is proportional to the absolute value of activity. In an attempt to capture the information about the state of the network, we propose averaging the sensitivity of all inactive neurons in a given layer of the network:

$$\widehat{\Omega}_l^{(\mathbf{x})} = \sum_{i \in \mathcal{I}_l} \frac{m_i}{N} \widehat{\boldsymbol{\omega}}_i, \tag{8}$$

where $\mathcal{I}_l$ is a set of all inactive neurons, $v_{li} \leq 0$ in layer $l$ of the network (excluding the output neurons), and $\widehat{\mathbf{c}}_i$ is the switched centre of neuron $i$; $m_i$ is the number of times neuron $i$ is on when the network is evaluated over the set of $N$ training samples. We refer to $\widehat{\Omega}_l^{(\mathbf{x})} \in \mathcal{R}^d$ as the *inactive state sensitivity* (Insens) of the network with respect to the input $\mathbf{x}$. The $m_i/N$ factor weights the importance of a neuron based on its activity over the training set; a neuron that is never on, $m_i = 0$, might represent an arbitrary pattern that has no useful information, while the neuron that is usually on, but not for the input for which $\widehat{\Omega}_l^{(\mathbf{x})} \in \mathcal{R}^d$ is computed, is taken to carry more weight.

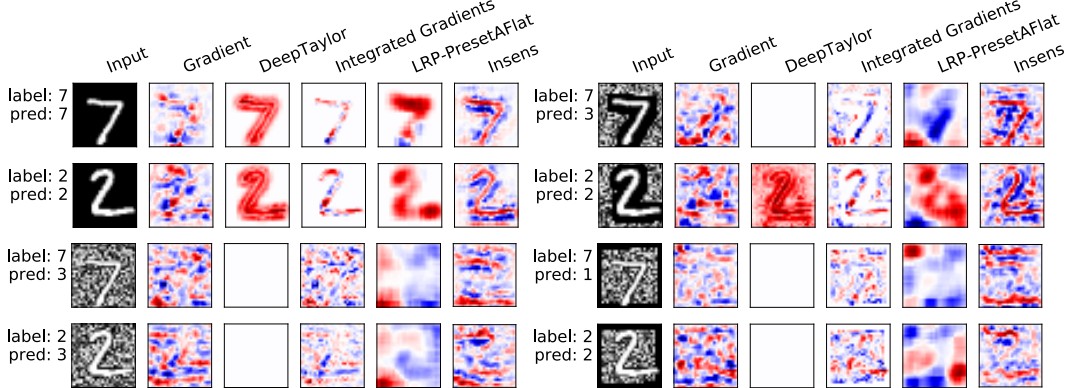

Figure 4: Visualisations for a set of interpretability methods of a single 2CONV neural network trained on the clean MNIST data in response to clean as well as noisy input; top-left shows clean MNIST, top-right noisy bordered MNIST, bottom-left noisy background MNIST and bottom-right noisy framed MNIST; the intensity of red and blue correspond respectively to the magnitude of the positive and negative values in the heatmaps.

## 4.1 EVALUATION

We evaluated Insens interpretability against plain gradient sensitivity, Deep Taylor decomposition (Montavon et al., 2017), Integrated gradients (Sundararajan et al., 2017) and Layerwise relevance propagation (Bach et al., 2015) as implemented by the iNNvestigate toolbox (Alber et al., 2018). For visualisations of the Insens-based interpretability we simply show $\widehat{\Omega}_l^{(\mathbf{x})}$ as a heatmap, with intensity of the red pixels relating the magnitude of its positive components, and the intensity of blue the magnitude of the negative component of the vector. Since the individual $\widehat{\omega}_i$ gives a weighted gradient with respect to the input, and thus the change vector that would make neuron $i$ less prone to become active, we take the average $\widehat{\Omega}_l^{(\mathbf{x})} \in \mathcal{R}^d$ to be indicative of the pattern in the input that is related to the stability of the network's state in layaer $l$ induced by $\mathbf{x}$. In all evaluations we examine visualisation of a 2CONV neural network (described in Appendix B)

First, we examine visualisations of $\widehat{\Omega}_{L-1}^{(\mathbf{x})} \in \mathcal{R}^d$, that is the inactive sensitivity over the penultimate layer of the network trained on the MNIST dataset (Lecun et al., 1998). We reason (and confirm below with saliency checks) that the last layer before the softmax output provides most information related to network's decision making out of all the hidden layers. Figure 4 shows visualisations from existing methods against the Insens heatmaps for different instances of the input and the same network. In the first instance we use clean MNIST input (on which test accuracy of 2CONV is over 99%). Note that Insens visualisation shows something that other methods hint at but do not show explicitly – that the network is sensitive to the black and white contrast of the digit and its outline. The red and blue areas in the Insens heatmaps suggest that respectively lightening and darkening these regions would make the current network state more stable (though in this case it is not possible to make black pixels darker nor white pixels any lighter). To verify the information provided by Insens, we added different types of noise to the images, also shown in Figure 4, and used them as input to the same network. We wanted to confirm that adding noise to the background, but not the digit outline, as suggested by Insens, does not impact performance. Indeed, testing with images that contain random gaussian noise in the background, but not within the 3-pixel outline of the digit (Figure 4 top-right) still gives 87% accuracy over 10000 test images. If we use images with random gaussian background everywhere (except the digit itself) or images with the same number of clean pixels as in the 3-pixel outline, but located around the frame of the image (Figure 4 bottom-left and bottom-right), the accuracy drops to 32% and 45% respectively. This confirms that Insens patterns for the penultimate layer provide meaningful information about the network's sensitivity. Also note that the other methods tend to lose the patterns of the digits in the visualisations over noisy images, whereas Insens still shows the digits, to which the network is still sensitive, albeit also being affected by the noise.

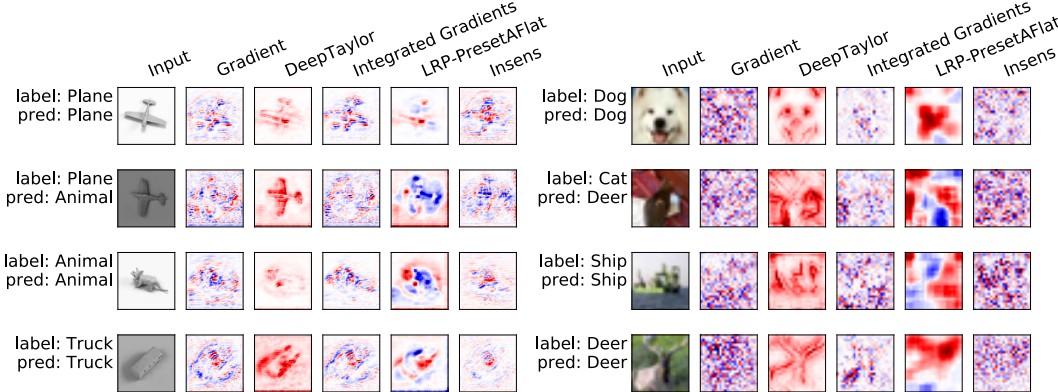

Figure 5: Visualisations for a set of interpretability methods on two 2CONV neural networks – one trained on the smallNORB dataset (left) and the other on the CIFAR10 dataset (right); the intensity of red and blue correspond respectively to the magnitude of the positive and negative values in the heatmaps.

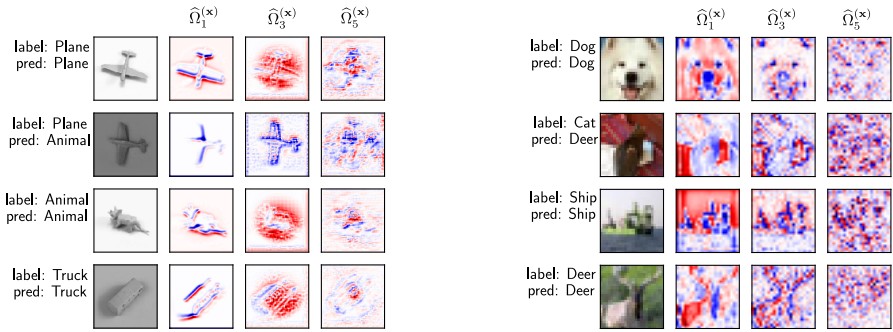

Figure 6: Visualisations for Insens $\widehat{\Omega}_l^{(\mathbf{x})}$ for layers l=1, 3, and 5 of the 2CONV network trained on smallNORB dataset (left) and CIFAR10 datasets (right); the intensity of red and blue correspond respectively to the magnitude of the positive and negative values in the heatmaps.

In Figure 5 we show evaluations of $\widehat{\Omega}_{L-1}^{(\mathbf{x})}$-based Insens and other methods for two additional 2CONV networks – one trained on smallNORB (LeCun et al.) and the other on CIFAR10 dataset (Krizhevsky, 2009) (for details of the training see Appendix B). Insens visualisation for the penultimate layer of smallNORB and CIFAR10 datasets might not seem all that different from other methods. However, with Insens we can examine the network state of a specific layer in the network, obtaining information about what the network pays attention to at different stages of its computation. In Figure 6 we show Insens visualisation for two convolutional layers ($l = 1$ and $l = 3$) as well as the penultimate, fully connected layer ($l = 5$) of the 2CONV network trained on smallNORB and CIFAR10. There are no Insens visualisations for the max-pooling layers, as these will contain only either active, or the arbitrary inactive chosen neurons by the max-pooling operation.

Finally, to provide an objective measure of the quality of Insens visualisations, we perform sanity and saliency checks as prescribed by Adebayo et al. (2018). In these tests we measure correlation between interpretability visualisations for different networks; in each instance a 2CONV network trained on a given dataset against an untrained (randomly initialised) network, and in the second instance the same dataset trained network against a network trained on identical input with randomly shuffled target labels. High correlation between visualisations from the same input between the trained and either randomly initialised or random label trained network suggests that an interpretability method is model agnostic, most likely showing an echo of the input. In Figure 7 we show average Spearman rank-order correlation coefficients between visualisations from 20 instances of independently trained 2CONV networks, for each using 100 randomly chosen images from the CIFAR10 dataset. In the first instance, we check the correlation of the penultimate layer-based Insens against other methods. Insens correlation is low, on par with the Gradient and Integrated Gradi-

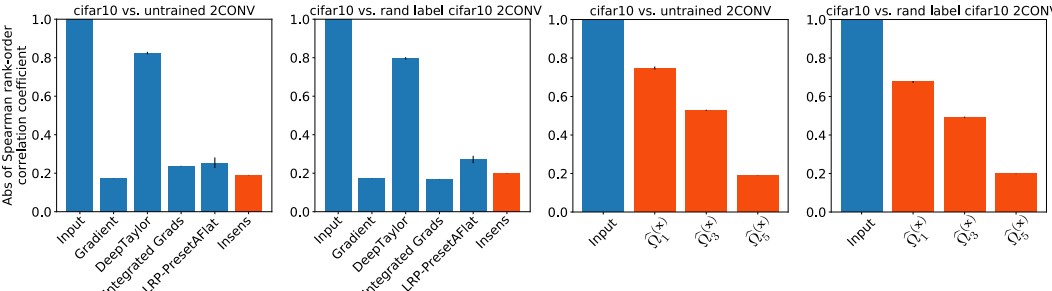

Figure 7: Mean spearman rank-order correlation between visualisations derived from a random sample of CIFAR-10 input for different methods including Insens $\widehat{\Omega}^{\mathbf{x}}_{L-1}$ between trained and randomly initialised 2CONV network (first left), between trained and random label CIFAR-10 trained 2CONV network (second left); the same correlation is shown over Insens based on different layers of the 2-CONV network (two figures on the right); Insens correlations are shown in red; vertical black lines show the variance.

ent methods. Note that the correlations between visualisation of the DeepTaylor method, which arguably provides the most visually striking visualisations, are very high, suggesting that most of the provided information is independent of the network's state. In the second instance we examine the correlations over Insens from different layers of the 2CONV architecture. It is very obvious that this correlation increases the closer the layer is to the input. We believe this is not an artefact of the Insens method, but an indication of the initial passing and then gradual filtering of information passing through the network.

## 5 CONCLUSION

The switched linear projection is an interpretation of the computation inside a ReLU network that distinguishes between the active and inactive parts of the deep neural network architecture. The active subnetwork tends to be a smaller subset of the deep network (see Appendix B.1 for details) and the linear projection it provides is somewhat arbitrary, in the sense that it does not matter what the orientation of the switched weight vector is, as long as it produces the desired output. Hence interpretability analysis based on the active subnetwork may not give the full picture of the patterns from the input that the network *relies* on in order to produce its computation.

Inactive state sensitivity (Insens), the proposed method for interpretability of ReLU networks, aggregates weighted sensitivity over a layer of inactive neurons. This sensitivity relates the gradient of the input that would potentially drive the activity of the inactive neurons further away from zero, thus corresponding to the patterns in a particular input that would keep the state of the particular layer stable and the output decision the same. As the name implies the method isolates the patterns in the input to which the network, in a sense, is *insensitive*. Our evaluations show that these patterns give a comprehensive picture of what and at what point of the computation the network *reacts* to patterns in a given input.

It is worth noting that it is possible to extend Insens by summing over all, active and inactive, neurons in Equation 8. In our experiments we found that inclusion of active neurons did not change the visualisations in a significant way, and we reasoned that the effect of these neurons is already accounted for in the computation of the switched linear projections on which the Insens evaluation is based on. However, it might be worth investigating in the future whether there are circumstances, datasets, or types of problems where inclusion of active state affects the sensitivity visualisations.

Since switched linear projections are just an interpretation of the computation inside a neural network, they may also become a useful tool for complexity analysis of deep networks. For instance, it might be possible to develop new regularisation methods based on switched weights, biases and centres of the neurons in the network. It remains to be investigated how the nature of the inactive subnetwork, and potential ways of manipulating it during training, would affect generalisation.

ACKNOWLEDGMENTS

We gratefully acknowledge the support of NVIDIA Corporation with the donation of the Titan X GPU used for this research.

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

## A  PROOF OF PROPOSITION 1

**Proposition 1** (Switched linear projections). *Let $\mathbf{x} \in \mathcal{R}^d$ be a vector of inputs, $\mathbf{w}_{li} \in \mathcal{R}^{U_{l-1}}$ the weight vector, and $b_{li} \in \mathcal{R}$ the bias of neuron $i$ in layer $l$ (with $U_{l-1}$ inputs from the previous layer). Let the activity of a neuron $i$ in layer $l$ be defined as:*

$$v_{li}(\mathbf{x}) = \Big(\ldots \sigma_r\big(\sigma_r(\mathbf{x}\mathbf{W}_1 + \mathbf{b}_1)\mathbf{W}_2 + \mathbf{b}_2\big)\ldots\Big)\mathbf{w}_{li} + b_{li}, \tag{1}$$

*where $\mathbf{W}_l = \begin{bmatrix}\mathbf{w}_{l1}^T & \ldots & \mathbf{w}_{lU_l}^T\end{bmatrix}$, $T$ denotes transpose, $\mathbf{b}_l = \begin{bmatrix}b_{l1} & \ldots & b_{lU_l}\end{bmatrix}$ and $\sigma_r(v) = \max(v, 0)$ is the ReLU activation function. If we define an input-dependent state of the network as*
$\mathbf{W}_l^{(\mathbf{x})} = \begin{bmatrix}\dot{\sigma}_r\big(v_{l1}(\mathbf{x})\big)\mathbf{w}_{l1}^T & \ldots & \dot{\sigma}_r\big(v_{lU_l}(\mathbf{x})\big)\mathbf{w}_{lU_l}^T\end{bmatrix}$ *and*
$b_l^{(\mathbf{x})} = \begin{bmatrix}\dot{\sigma}_r(v_{l1}(\mathbf{x}))b_{l1} & \ldots & \dot{\sigma}_r(v_{lU_l}(\mathbf{x}))b_{lU_l}\end{bmatrix}$,
*where $\dot{\sigma}_r(v) = \frac{d\sigma_r(v)}{dv}$, then for*

$\widehat{\mathbf{w}}_{li}^T(\mathbf{x}) = \mathbf{W}_1^{(\mathbf{x})}\mathbf{W}_2^{(\mathbf{x})}\ldots\mathbf{W}_{l-1}^{(\mathbf{x})}\mathbf{w}_{li}^T$ *and*

$\widehat{b}_{li}(\mathbf{x}) = \mathbf{b}_1^{(\mathbf{x})}\mathbf{W}_2^{(\mathbf{x})}\ldots\mathbf{W}_{l-1}^{(\mathbf{x})}\mathbf{w}_{li}^T + \mathbf{b}_2^{(\mathbf{x})}\mathbf{W}_3^{(\mathbf{x})}\ldots\mathbf{W}_{l-1}^{(\mathbf{x})}\mathbf{w}_{li}^T + \ldots + \mathbf{b}_{l-1}\mathbf{w}_{li}^T + b_{li}$, *we have*

$$v_{li}(\mathbf{x}) = \mathbf{x}\widehat{\mathbf{w}}_{li}^T(\mathbf{x}) + \widehat{b}_{li}(\mathbf{x}). \tag{2}$$

*Proof.* By definition from Equation 1, the activity of neuron $i$ in layer $l$ is

$$v_{li}(\mathbf{x}) = \sum_{j=1}^{U_l} \sigma_r\Big(v_{l-1j}(\mathbf{x})\Big)w_{lij} + b_{li}, \tag{9}$$

where $w_{lij}$ is the weight on the connection between neuron $j$ of layer $l-1$ and neuron $i$ of layer $l$, and $b_{li}$ is the bias of neuron $i$ in layer $l$.

Since $\sigma_r(v) = \begin{cases} v & v > 0, \\ 0 & \text{otherwise,} \end{cases}$ and $\dot{\sigma}_r(v) = \frac{d\sigma_r(v)}{dv} = \begin{cases} 1 & v > 0 \\ 0 & \text{otherwise,} \end{cases}$ we have

$$\sigma_r\Big(v_{l-1j}(\mathbf{x})\Big)w_{lij} = \begin{cases} v_{l-1j}(\mathbf{x})w_{lij} & v > 0, \\ 0 & \text{otherwise,} \end{cases}$$

and thus

$$\sigma_r\Big(v_{l-1k}(\mathbf{x})\Big)w_{lik} = v_{l-1k}(\mathbf{x})\dot{\sigma}_r\Big(v_{l-1k}(\mathbf{x})\Big)w_{lik}.$$

As a result

$$v_{li}(\mathbf{x}) = \sum_{j=1}^{U_l} v_{l-1j}\dot{\sigma}_r\Big(v_{l-1j}(\mathbf{x})\Big)w_{lij} + b_{li} = \sum_{j\in\mathcal{A}} v_{l-1j}w_{lij} + b_{li} \tag{10}$$

Table 1: 2CONV network (conv = convolution; fc = fully connected)

| Layer | Type | Filters/ neurons | Window | Stride | Activation |
|-------|------|------------------|--------|--------|------------|
| 1 | conv | 32 | 5x5 | 1x1 | relu |
| 2 | maxpool | - | 2x2 | 2x2 | relu |
| 3 | conv | 64 | 5x5 | 1x1 | relu |
| 4 | maxpool | - | 2x2 | 2x2 | relu |
| 5 | fc | 512 | - | - | relu |
| 5 | fc | $K$ | - | - | - |

where $\mathcal{A}$ is the set of neurons with activity $v > 0$, the active neurons. Substituting the expression for activity from Equation 10 into its recursive definition in Equation 9, where $v_{0i}(\mathbf{x}) = x_i$, reveals that the overall computation is a series of linear transformations of $\mathbf{x}$ equivalent to a single linear transformation

$$v_{li}(\mathbf{x}) = \mathbf{x}\widehat{\mathbf{w}}_{li}^T(\mathbf{x}) + \widehat{b}_{li}(\mathbf{x}).$$

where $\widehat{\mathbf{w}}_{li}^T(\mathbf{x}) = \mathbf{W}_1^{(\mathbf{x})}\mathbf{W}_2^{(\mathbf{x})}\ldots\mathbf{W}_{l-1}^{(\mathbf{x})}\mathbf{w}_{li}^T$ and

$$\widehat{b}_{li}(\mathbf{x}) = \mathbf{b}_1^{(\mathbf{x})}\mathbf{W}_2^{(\mathbf{x})}\ldots\mathbf{W}_{l-1}^{(\mathbf{x})}\mathbf{w}_{li}^T + \mathbf{b}_2^{(\mathbf{x})}\mathbf{W}_3^{(\mathbf{x})}\ldots\mathbf{W}_{l-1}^{(\mathbf{x})}\mathbf{w}_{li}^T + \ldots + \mathbf{b}_{l-1}\mathbf{w}_{li}^T + b_{li}. \qquad \square$$

## B    2CONV ARCHITECTURE

This network consists of two convolutional layers each followed by a maxpool layer that down-samples the feature map, followed by a single fully connected layer of 512 neurons and $K$ output neurons, where $K$ is the number of classes in the dataset of interest (see Table 1 for details). For training this network we used Adam optimiser minimising the softmax cross-entropy without regularisation. In all training a portion of the training set was set aside for validation for early stopping.

For the first, MNIST evaluation, the network was trained on cleaned MNIST data to test accuracy of 99.4%. The network was then tested on several variant of the MNIST dataset: MNIST Gauss images were given random Gaussian background (test accuracy of 31.7%), MNIST outline images with same Gaussian background except for outline within 3 pixels of the digit (test accuracy of 86.0%), and MNIST frame with Gaussian background and the same number of clean pixels as in the MNIST outline version of a given image around the frame (test accuracy of 44.9%).

For the second, smallNORB evaluation, the network was trained four times on variants of the dataset differing in the encoding of the images: smallNORB with grayscale intensity given by value between 0 and 1 for black to white pixel respectively, smallNORB neg with grayscale intensity between 0 and 1 for white to black pixel, smallNORB mean with grayscale between -1 and +1 for black to white pixel and smallNORB neg mean with grayscale between -1 and 1 for white to black pixel. The classifciation accuracy on these four networks tested on 24300 image of each dataset was respectively 84.3%, 87.2%, 81.4% and 83.6%.

For the third, CIFAR-10 evaluation, a single network was trained on augmented (by random contrast adjustment and crop to a 24x24 size) images of the CIFAR-10 dataset giving test accuracy of 81.0%.

### B.1    NUMBER OF ACTIVE NUEURONS

Table 2 shows the average percentage of neurons that are active during input-output mappings for the trained networks evaluated in this paper.

## C    MULTIPLE OBJECTS OF INTEREST AND ADVERSARIAL EXAMPLES

We evaluate Insens against other methods on a classification task over images with two objects – one to of interest and the another to be ignored. For this experiment we created a dataset where the

Table 2: Average percentage of active neurons in a neural network

| Dataset | Network | Total # neurons | Average % active neurons |
|---------|---------|-----------------|--------------------------|
| MNIST | 2CONV | 38144 | 17% |
| MNIST gauss | 2CONV | 38144 | 25% |
| MNIST outline | 2CONV | 38144 | 27% |
| MNIST border | 2CONV | 38144 | 23% |
| smallNORB | 2CONV | 442880 | 20% |
| CIFAR-10 | 2CONV | 28160 | 19% |

input consists of an even and odd MNIST digit concatenated in random order. The training label identifies the even digit, making it the object of interest. We call this dataset MNISTdbl. Figure 8 shows heatmap visualisations for penultimate-layer Insens and other methods for a 2CONV network trained on this dataset to achieve 99.5% test accuracy. All the methods seem to emphasise the object of interest mostly by making the even digit brighter and more prominent in the visualisation. LRP visualisation seems to also differentiate the two objects by colour – with the object of interest highlighted in red and the other in blue.

For a deeper evaluation of these interpretability methods, we obtain visualisations from the same 2CONV network (trained on the MNISTdbl images) using different types of adversarial examples: with only the object to ignore present (Figure 9), two objects of interest in a single image (Figure 10) and two objects to ignore in a single image (Figure 11). Through the same red/blue colour highlighting scheme as in the previous example, LRP-based visualisations tend to communicate rather well which objects in the image are those of interest and which ones are the ones the network has learned to ignore. It does not provide too much explanation of the predictions that the network outputs for these examples. DeepTaylor and Integrated Gradients methods highlight the digit used for prediction in the two even digits case. Insens visualisation might be providing information about the decisions with respect to the output predictions, but other than the fact that the objects of interest tend to get better outlines, it is hard to say what the nature of these decisions might be. Most intuitive expectation is for the predicted digit to show up in the visualisation. However, this is making an assumption about the way the network makes its decisions, and, as other adversarial examples show, it's not necessarily the case that the network *perceives* the shape of the shape of the digits like we do.

Finally, we include an evaluation of the penultimate layer Insens and comparison to other methods on gradient-based adversarial examples, where a correctly classified image has been modified in the direction of the input gradient obtained from a trained 2CONV network to increase the activity on the output corresponding to the wrong label. Figure 12 shows visualisations from different interpretability methods over different adversarial examples derived from the same image for the MNIST and smallNORB trained 2CONV networks. Figure 13 shows the difference between the visualisations from the real and adversarial input. Once again, the predicted digit does not show up in any of the visualisations, though again, Insens does indicate subtle changes, especially in the MNIST examples, with parts of the digits missing and some extra bits showing up.

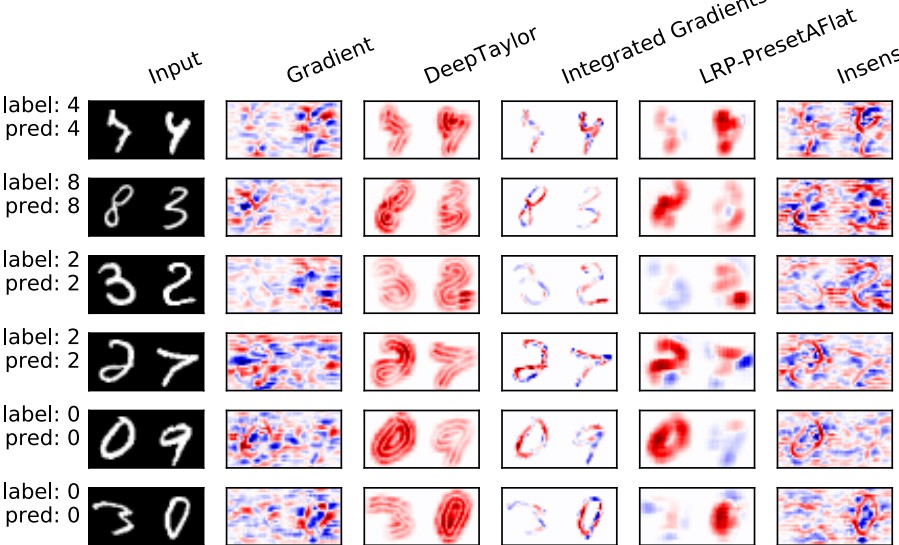

Figure 8: Visualisations of the 2CONV architecture trained on MNISTdbl dataset, where input is a composite of an odd and even digit from the MNIST database and the target label corresponds to the even digit; input images shown are test images from the MNISTdbl dataset.

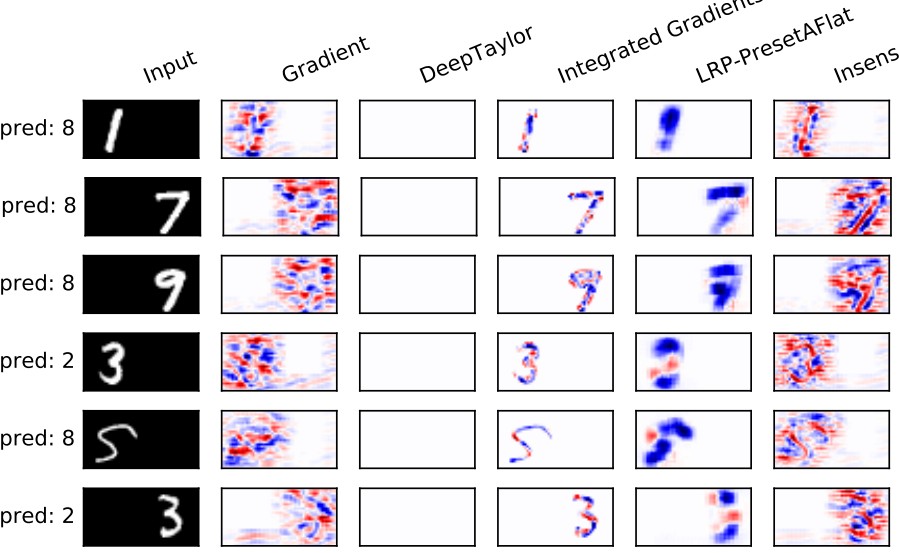

Figure 9: Visualisations of the 2CONV architecture trained on MNISTdbl dataset, where input is a composite of an odd and even digit from the MNIST database and the target label corresponds to the even digit; input images shown are adversarial examples where only the odd digit is present.

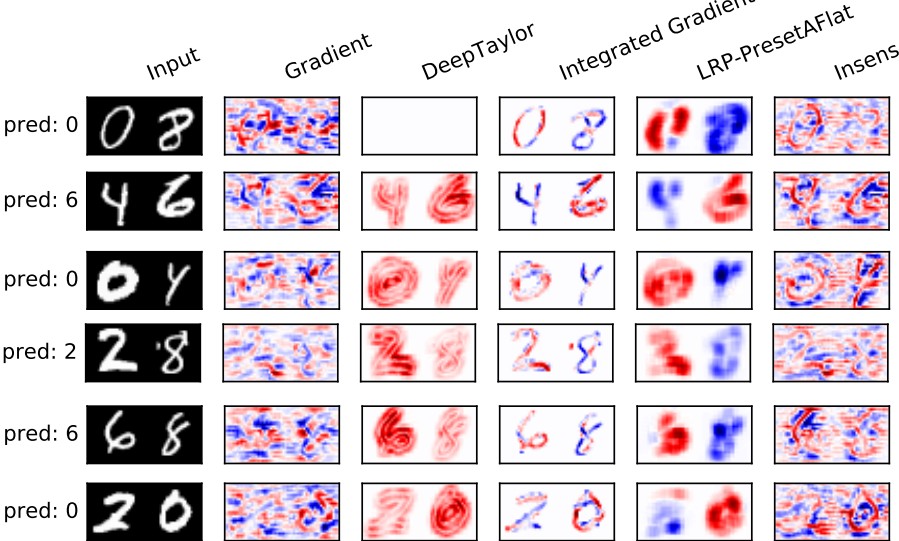

Figure 10: Visualisations of the 2CONV architecture trained on MNISTdbl dataset, where input is a composite of an odd and even digit from the MNIST database and the target label corresponds to the even digit; input images shown are adversarial examples with two even digits.

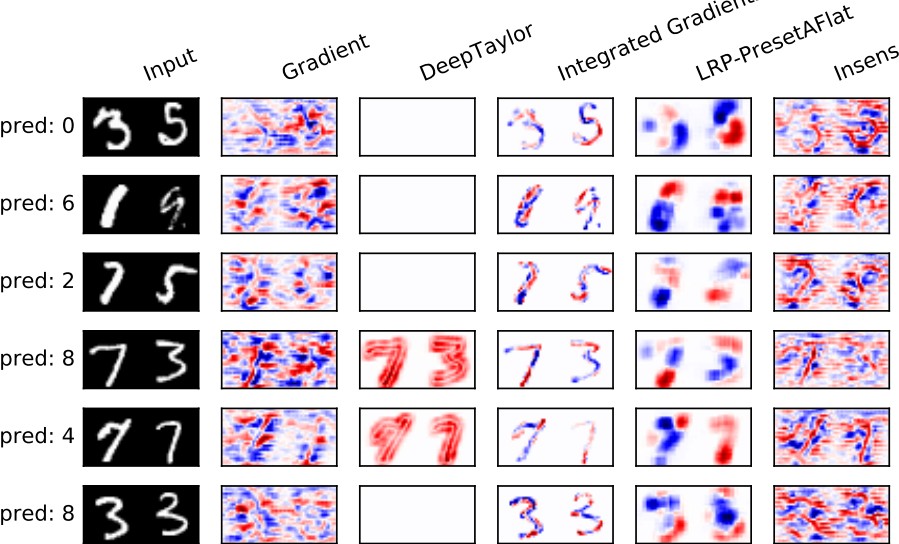

Figure 11: Visualisations of the 2CONV architecture trained on MNISTdbl dataset, where input is a composite of an odd and even digit from the MNIST database and the target label corresponds to the even digit; input images shown are adversarial examples with two odd digits.

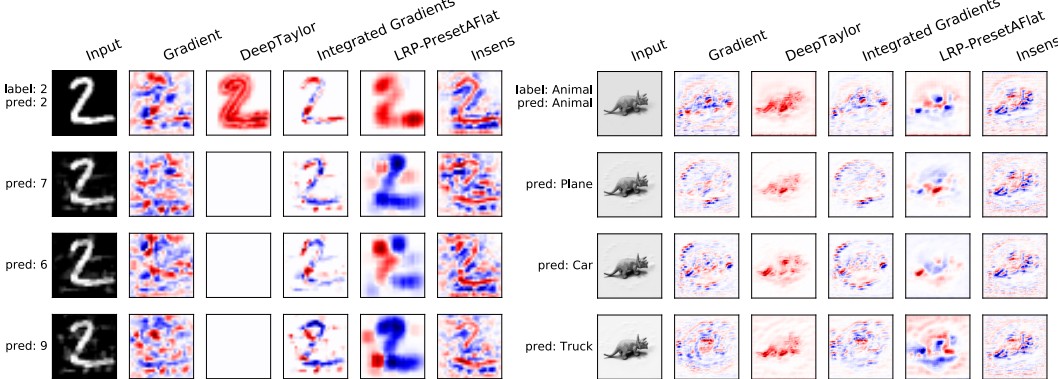

Figure 12: Visualisations for a set of interpretability methods of a single 2CONV neural network trained on the MNIST dataset (left) and smallNORB dataset (right) in response to advesrarial examples; the intensity of red and blue correspond respectively to the magnitude of the positive and negative values in the heatmaps.

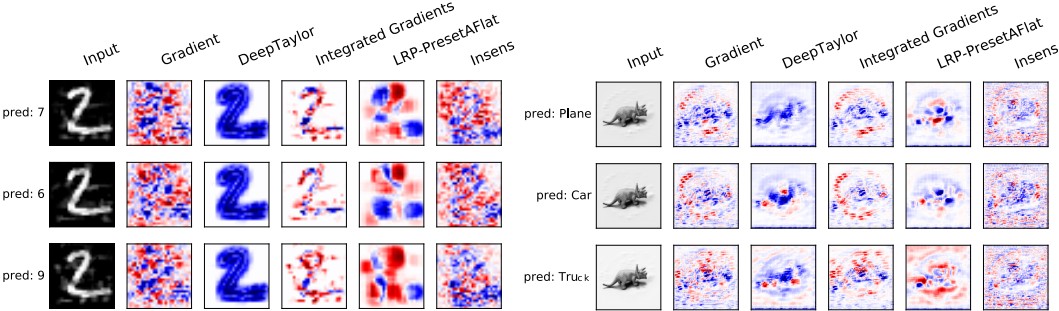

Figure 13: Visualisations of the difference between the visualisations from Figure 12; the difference shown is between visualisation based on the image from the dataset (top row Figure 12) and respectively each adversarial example visualisation (second, thirds and fourth rows of Figure 12; the intensity of red and blue correspond respectively to the magnitude of the positive and negative differences of values in the heatmaps.

