# OpenReview forum: "Switched linear projections and inactive state sensitivity for deep neural network interpretability"
_ICLR.cc/2020/Conference — Reject_

### Official Review · AnonReviewer1 · 2019-10-20
**Official Blind Review #1**

**Rating:** 6

**Review:**

Notes:

  -Goal is to study the "active subnetwork" of Relu based networks for interpretability.

  -The question of interpretation seems rather thorny.

  -In Figure 4, the result for Insens seem alright, although it's weird that the data is just mnist digit / noise.  I feel like something with multiple objects would make it much clearer if there is an actual improvement?  For example on Figure 5 I'm not really sure if Insens is better.  The results often look worse to me than "DeepTaylor", especially on CIFAR10.

Review: This paper proposes to improve the interpretation of relu based networks by considering the "inactive network" which could potentially become activated by local perturbations instead of just considering the active part of the network (which is locally linear).  I think this is a step in the right direction for the interpretation of relu based networks, although the results are somewhat borderline.

Additionally the tasks could be much better, to show situations where an object is present but which is not related to the labels.  This would provide a much clearer test of the model's capabilities.



**Experience Assessment:**

I have published in this field for several years.

**Review Assessment: Checking Correctness Of Derivations And Theory:**

I carefully checked the derivations and theory.

**Review Assessment: Checking Correctness Of Experiments:**

I carefully checked the experiments.

**Review Assessment: Thoroughness In Paper Reading:**

I read the paper thoroughly.

---

> ### Author Response · Authors · 2019-11-11
> **Review #1 response**
>
> Thank you for your review and suggestions.
>
> As the reviewer pointed out, in the original paper the Insens visualisations were similar to DeepTaylor one, on occasion perhaps not being even equally good.  Prompted by another review, we carried out saliency and sensitivity analysis which show that both the original Insens and DeepTaylor produced similar visualisations for the same input regardless of the model it passed through. However, after redefining Insens to relate information of the inactive neuron sensitivity over individual layers of a network (in the latest edition of the paper), we created a method for layered interpretability, from the early stages where neurons "look at the entire input", to "decision making" patterns of the penultimate layer of the network.
>
> The suggestion of evaluation over images with objects that need to be ignored is quite a nice idea.  When working on new experiments for the layered version of Insens, we did create a simple MNIST dataset with two different digits in the input image and one label.  In the first instance, the digit to be labeled was always on the left-hand side...in which case the task was trivial - the network easily learns to ignore the other side of the image, and the non-label digits are not highlighted in all visualisation methods (or are a bit fainter in case of DeepTaylor).  Then we tried a task where the each image contained randomly ordered even and odd number, with the even number labelled.  This network didn't train well over the 2CONV architecture (poor test accuracy).  The option of trying a bigger network is not appealing, since if the visualisation shows both digits as being important, it is not clear whether the visualisation is wrong. After all, it's probably possible for the network to "memorise" all combinations of possible digits (at least in this test) and thus show both as important.  Hence, we decided not to pursue this experiment any further, especially since at best it could only make it to the appendix of the paper.

---

> > ### Comment · AnonReviewer1 · 2019-11-14
> > **Thanks.**
> >
> > I still think that the tasks could be much better where an object present is not related to the labels etc. This would provide a much clearer test of the model's capabilities.

---

> > > ### Author Response · Authors · 2019-11-15
> > > **Multiple objects**
> > >
> > > After this comment, we went back to work on the double MNIST digit experiment...and spotted a small bug in the labelling of the composite digits, which prevented the network from training well.  Afterwards we were able to train the 2CONV architecture to a very good test accuracy on input made up of composites of two MNIST images - one of an even and another of an odd digit and the label indicating the even digit.  We also came up with some additional test scenarios using adversarial images where the object of interest (the even digit) is missing or two different even digits are present.  It was impossible to fit it all into the main body of the paper, but we have included visualisations from this set of experiment in Appendix C of the latest revision of the paper.

---

> > > > ### Comment · AnonReviewer1 · 2019-11-15
> > > > **Thanks.**
> > > >
> > > > "Insens visualisation might be providing information about the decisions with respect to the output predictions, but other than the fact that the objects of interest tend to get better outlines, it is hard to say what the nature of these decisions might be. Most intuitive expectation is for the predicted digit to show up in the visualisation. However, this
> > > > is making an assumption about the way the network makes its decisions, and, as other adversarial examples show, it’s not necessarily the case that the network perceives the shape of the shape of the digits like we do."
> > > >
> > > > I actually agree with this. Its very hard to actually say what the nature of these decisions are. So, I'm not sure, if it really helps me make a better judgement of the paper.
> > > >
> > > > One way would be to use something like https://arxiv.org/abs/1610.01644, and probe the representations for different labels at each layer, and see what kind of information each layer is learning. But I understand its a bit late as rebuttal period is going to end now.

---

> > > > > ### Author Response · Authors · 2019-11-15
> > > > > **Classifier probes**
> > > > >
> > > > > Indeed, looking at the sanity check in  Figure 7 of Insesn over different layers we do wonder about possibility of using some means of separating the decision impacting and input-reflecting information coming through Insens  in the early layers.  We are not able to incorporate classifier probes analysis into this paper, but thank you for pointing us towards it.

---

### Official Review · AnonReviewer3 · 2019-10-23
**Official Blind Review #3**

**Rating:** 6

**Review:**

The work basically introduced a new way of looking at interpretability; instead of focusing on the source of activations in the network for a given input image, focus on the source of stability (non-active) neurons (in a ReLU network). The work starts by proving (although it is trivial) that in a ReLU (more generally any piece-wise linear) network, for a given input image, there is a locally linear relationship between a given neuron's activation and the image: v= w^T x + b. As the authors correctly mention, focusing on 'w' as the sensitivity analysis is basically the vanilla gradient method. The contribution, however, is focusing on the projection of bias and the introduced notion of 'centre'. With this provided notion, one can focus on the deactivated neurons in the network and how each input pixel is responsible for it. In other words, unlike previous work that focuses on the activation map, the authors correctly refer to the deactivated neurons as another source of the network's prediction.

I'm have reasons for both accepting and rejecting this work.  The work provides a new perspective and asks a very interesting question. The introduced method, although quite simple and trivial, is useful and the authors do a very good job of making valid and reasonable claims about their work's contribution and how it connects to the existing literature. The main drawback of the paper, however, is whether the contributions are enough for this venue. The paper does not convince me that the introduced method would result in better interpretability of deep networks compared to what is already there. Another minor (or for some people in the field major) issue is the experimental setup.  All of the experiments are focused on subjective examples and no objective measure of the introduced method is provided (and the field has many of those objective measures). Providing a few examples of the method in comparison with other methods is not sufficient. Anyhow, the experiment where they prove the usefulness of the method by adding background noise is interesting. I would personally suggest the authors to expand this experiment to testing the method's sanity using the sanity measures provided in previous work: https://arxiv.org/abs/1810.03292 The claims made about the results on smallNORB can be controversial as the authors interpret their method's flipping of importance to be the reality of what's happening in the network and the other method's focus on the edges as false; this is not clear to be true. My score would be subject to change if better experimental results are provided (and the other way round).

A few suggestions and questions:
* One very important issue with the method is that it considers all of the inactive neurons. We know that a substantial percentage of inactive neurons are just dead neurons the stability of which does not matter. How would the method address the issue?
* There definitely needs to be an objective measure of the introduced method's performance compared to previous work.
* The work seems very related to DeepLIFT while there is no mentioning of it.
* I'm not a fan, but adding results on a SOTA ImageNET paper always helps with making the experiments section crisper.
* The authors claim that even small perturbations will change the activation pattern. This is not a small claim and is definitely in need of more evidence.

**Experience Assessment:**

I have published in this field for several years.

**Review Assessment: Checking Correctness Of Derivations And Theory:**

I carefully checked the derivations and theory.

**Review Assessment: Checking Correctness Of Experiments:**

I carefully checked the experiments.

**Review Assessment: Thoroughness In Paper Reading:**

I read the paper thoroughly.

---

> ### Author Response · Authors · 2019-11-11
> **Review #3 response**
>
> Thank you for very insightful review.  The time taken to respond is a consequence of some soul searching and a decent amount of work we had to undergo in order to address the request for objective evaluation of the proposed method.
>
> After quick test based on Abedayo et al's objective measures of visualisation, it became immediately apparent that the correlation between Insens visualisations from the trained, random weights and random label networks, though lower than DeepTaylor's, is still quite high.  However, after an investigation, we realised that if we compute Insens over the inactive neurons of a given layer (rather than over the whole network), then these correlations vary from layer to layer.  While this required a slight change of definition of Insens,  visualisations based on the penultimate layer of the network were still very good.  This new definition also created the potential for visualisation of the inactive state layer by layer.  We feel that these changes do not detract from the main argument of the paper, which promotes the use of inactive state for interpretability.  Hence, we’ve decided to change the evaluation a bit and add the saliency tests.  The new version of the paper has been uploaded.
>
> With respect to completely dead neurons, initially we thought these could be simply taken out of the network before the Insens computation.  But thinking about importance of the neurons that are off for most of the time (when tracking their activity over all training samples) as opposed to those that are inactive half of the time, we decided to supplement the Insens sum with a weighting factor.  The factor m_i/N multiplies the insensitivity vector by the ratio of the number of times the neuron in question is active (over the training data) , m_i, over the number of training examples, N.  For neurons that are always off m_i/N=0, for those that are almost always on, but inactive for a given input in Insens, m_i/N -> 1.
>
> We have included an analogy to DeepLIFT’s reference input when explaining the concept of neuron’s centre.
>
> The main reason why we can’t at this time include the state of the art ImageNET visualisations is because our computation of Insens is (at this moment) not very efficient.   In VGG-16, there are, on average, 50% inactive neurons for a given input - that’s many millions of inactive neurons.  The easiest way to compute a switched vector is to take the derivative of neuron activity with respect to the network input, and then find \hat{b} by subtracting \hat{w}x from v.  Since our implementation is in Tensorflow, it would seem easy, but Tensorflow creates separate graph for each derivative of activity with respect to the input.  While it does have a single operation for computing sum of derivatives, this would give us a sum of \hat{w}, whereas for Insens we need the sum of \hat{w}-\hat{c} (and we can't calculated these sums separately since \hat{c}=v\hat{w}/(\hat{w}^t\hat{w}) is not independent of \hat{w}).  It should be possible to add an operation to Tensorflow that does what we need in one graph, but at this point we don’t have the Tensorflow know-how on how to do this in such a short time.
>
> In relation to our claim that even small perturbations will change the activation pattern, this is based on some sporadic experimental evidence, which is not ready for publication and quite possibly would qualify as a different paper altogether.  We removed this statement from our conclusion and toned down the claims of Insens being better than active-based visualisation. Nevertheless we feel that the idea of using inactive state for interpretability and Insens are still strong contributions that supplement existing methods for interpretability.

---

> > ### Comment · AnonReviewer3 · 2019-11-13
> > **Rebuttal**
> >
> > Thanks for the detailed response and clarification. Providing the new results (specifically the sanity check results) is quite impressive. Just to make sure, the new changes in the method are two-fold: 1- adding the weighting factor in the summation 2-layer-specific interpretations ?
> > PS: There are reference typos on pages 5, 7.

---

> > > ### Author Response · Authors · 2019-11-13
> > > **New changes**
> > >
> > > That is correct,  these are the two changes to the method.  We'll fix the typos in the next revision of the paper, once the results over 20 evaluations of sanity checks are ready.

---

### Official Review · AnonReviewer4 · 2019-11-07
**Official Blind Review #2**

**Rating:** 3

**Review:**

This manuscript introduces a novel method to explain activities of ReLU-based deep networks by constructing a linear subnetwork which only contains neurons activated by the input. The status of each neuron can be obtained given any input sample. Moreover, the author applies the notion of “neuron’s center”, which is a neutral data point that is similar to actual input x, but with differences in particular objects to cause f(x) be positive. The activity of each neuron can be decomposed into the attribution of each input pixel, and this decomposition can also be used to measure the contribution of each pixel to the network stability. Overall, the proposed methodology is intuitive and distinctive to the state-of-the-art interpretability methods.

However, the application constraint on the ReLU-based deep neural network prevents this method from being a model agnostic approach: the problem formulation would be much different if other non-linear activation functions are used. Although the experiment part visualizes the superiority of switched linear projections over other prevalent approaches, the evaluations contain mostly subjective assessment and the arguments are monotonous. I would suggest adding more experiments with quantitative analysis, or mathematically demonstrate why the proposed method is better than, say purely gradient-based method, in the linear case. In addition, additional experiments on a broader set of input data (e.g., tabular, text) could avoid the evaluations look cherry-pick.

Minor issues:

1. In figure 2, I think it would be better to write down explicitly the connections between v, \hat{b} and \hat{w} for each neuron given any input. Just seeing v and \hat{b} on top of each subfigure is a bit confusing.
2. I spent a long time to understand the "neuron’s center" concept, it might be better to add some background or mathematical formulation.
3. In figure 4, when the digits get misclassified, the Insens explanation should highlight the patterns of wrongly predicted digits, but the patterns of neurons' inactive state sensitivity still look like the correct digits.
4. It would also be interesting to show how the Insens explanation would change when the input is under various kinds of adversarial attacks rather than adding simple Gaussian noise.


**Experience Assessment:**

I have read many papers in this area.

**Review Assessment: Checking Correctness Of Derivations And Theory:**

I assessed the sensibility of the derivations and theory.

**Review Assessment: Checking Correctness Of Experiments:**

I carefully checked the experiments.

**Review Assessment: Thoroughness In Paper Reading:**

I read the paper thoroughly.

---

> ### Author Response · Authors · 2019-11-11
> **Review #2 response**
>
> Thank you for taking time to review our paper.
>
> Indeed, Insens in not model agnostic, but the driving idea of this work is that inactive neurons carry information about what is happening inside the network.  For non-ReLU networks there is no notion of inactive neurons as it would be hard to find, in a sigmoid network for instance, a neuron outputting exactly zero. The non-linearity at exactly 0 is a unique feature of ReLU networks that allows separation of the architecture into active and inactive sub-networks - a feature that can be exploited for the analysis presented in the paper.  In fact, the switched linear projection itself is generalisable to networks with non-ReLU functions - we hint at this in the paper when we state that switched projection is equivalent to derivative of activity with respect to the input.  So, \hat{w}=dv/dx, and therefore \hat{b}=v-\hat{w}x, which can be computed for a neuron from a network of any activation function (as long as the function has a first derivative).  However, since we were specifically interested in the notion of separating the non-active subnetwork from the rest of the architecture, we decided, in the interest of readability, to limit the paper to ReLU networks only.  We believe that this is still a worthy and useful contribution on the account of prevalence of the ReLU networks in deep learning applications.
>
> Following other reviewer's suggestion, we did include an objective and quantitative analysis of Insens based on salience and sanity checks of Abedayo et al's in the latest version of the paper.  As it turned out, the quality visualisations of the original definition of Insens , and the DeepTaylor method, were so good mostly because they echoed the input rather than what the model was doing.  The new definition of the layered Insens, considers the inactive states of neurons of one layer at a time. The visualisations from the penultimate layer are still of decent quality, but now definitely not simply echoing the input.  At the same time, we can use the new layered definition to track the inactive state through different layers of the network.
>
> We agree that more experiments on a broader set of input data is always a great idea, but the space is limited, and we felt that image-based visualisation are sufficient to present our novel method.  However, as the reviewer suggested, we did add visualisations for adversarial examples in the appendix.
>
> In Figure 4, where the noisy digit gets misclassified, indeed the visualisation does not necessarily produce an image of the wrong digit.  But this assumes that the only way for a network to misclassify a digit is to mistake it for another one.  It's important to keep in mind that, given softmax output, the network divides the space of all images into digits.  As a result, there are many images that have nothing to do with numbers, yet the network will classify as one of the digits.  Thus, as in the case of the adversarial examples included in the appendix, the network does not necessarily "see" the wrong digit when making the wrong prediction.  Instead, it sees the noise that happens to produce a particular label.  We believe Insens visualisations highlight that noise, but it's hard for us to see patterns in it, since the network is not reacting to a shape-based pattern.

---

### Official Review · AnonReviewer7 · 2019-11-12
**Official Blind Review #7**

**Rating:** 1

**Review:**

This paper proposes a method to capture patterns of “off” neurons using a newly proposed metric. While the authors have considered only linear networks, the setup is still relevant because how often these networks can give meaningful results, and can possibly pave the way for future research into more general networks.

Pros: The idea itself is interesting, the related works are discussed well, and MNIST experiments are very interesting.

Cons/comments : The writing needs a lot of improvement if to be considered for a top venue like iclr.  Other than the MNIST experiments, which show and indicate the importance of relative contrast and boundary, I am not sure how other experiments are meaningful. CIFAR and smallnorb experiments are merely presented, without discussions on what the interpretation shows or helps over the existing methods. Infact, the other methods seem to capture a lot more information than the proposed method. I would suggest adding more discussions and more experiments that show interpretation that this method/metric helps with to make this work stronger.

Have the authors considered  the metric to consider “on” neurons instead of “off”neurons ? Is it possible to have a general metric that combines the two in some way ? Intuitively, its unclear to me why only the off patterns can help (except in specific cases as shown in MNIST experiments).

 “and thus is responsible for the activity vi” – This is unclear to me. I understand the projection part though, but I cant make sense of this statement.

“interpretation and interpretability ” in the introduction – the writing is too informal. Making use of italicized phrases like “switched linear projection” does not help with the understanding at all, especially because “switched” is defined after using the term atleast thrice.

Confirmation bias <-> “information we want to get”.

The same issue right after eq 6. “Reference subtracted from ….” where the first word is italicized to probably imply some intuitive explanation, but for someone not familiar with what reference is just tends to confuse the reader.
Please fix missing references.

Eq 7 seems written incorrectly, with the where “v= …”. Please fix.

What is the variance of saliency checks ? In other words, if the experiment of 100 random samples is repeated (say) 20 times, how different are the corresponding coefficients across these repeated runs ?

Figure 3 is waste of space (move to appendix?)

I might be splitting the hairs but Theorem 1 does not warrant a theorem. The result/proof is too straightforward to be a theorem and is already known in some form in the folklore.



**Experience Assessment:**

I have published one or two papers in this area.

**Review Assessment: Checking Correctness Of Derivations And Theory:**

I carefully checked the derivations and theory.

**Review Assessment: Checking Correctness Of Experiments:**

I carefully checked the experiments.

**Review Assessment: Thoroughness In Paper Reading:**

I read the paper thoroughly.

---

> ### Author Response · Authors · 2019-11-13
> **Review #7 response**
>
> Thank you for reviewing our work.
>
> The objective of Figure 5 is to provide a comparison of the penultimate-layer Insens, as well as layer-by-layer Insens from Figure 6, against other methods on more challenging datasets than MNIST.  We do make a small comment (in a segue to layer by layer visualisations and Figure 6) conceding that the penultimate-layer Insens does not necessarily provide significantly improved visualisations. However, what makes up for that is the fact that Insens can show what each layer is doing in the neural network.  This seems like a significant feature that other methods lack.  Also, the statement that other methods "seem to capture a lot more information" needs to be stacked against the saliency and sanity checks provided in Figure 7.  Sure, Deep Taylor looks like it's showing us more, but it turns out it's just showing us back input, and not what the network is doing.  The problem with these visualisations is that we assume they "outline" the object of interest in the image - but that's making an assumption that these networks recognise by shape.  It's been shown that in fact they seem to "perceive" by texture rather than shape (Geirhos.etal, ICLR 2019).
>
> Yes, we have considered including "on" neurons.  Our initial version of Insens (before it was called Insens) was averaging \hat{w}-\hat{c} from all the neurons.  We found that active neurons don't change the visualisations in a significant way.  We considered whether to propose a more general version of Insens, but decided to focus on the message of importance of the "off" neurons, which hasn't been considered before.  One possible objection to inclusion of active neurons is that their contributions are accounted in the switched projection, whereas the impact of the switched off neuron terminates at that neuron.  There is a potential for more generic future methods based on \hat{w}-\hat{c}, but we feel that pointing out the significance of the "off" neurons is a important contribution that would be of interest to many readers, who later on may go on to use this approach in different and perhaps even improved ways.  Having said all that, we concede that an explicit explanation to that effect should be made in the paper, since inclusion of active neurons is something that perceptive readers might wonder about.
>
> The bar height in Figure 7 gives the mean of correlation over 100 samples and there is a bar (perceptible only in DeepTaylor and LRP) that shows the variance.  We can increase the sample size (is the reviewer suggesting we need to do it over 2000 samples?).  We are happy to add something to the caption explaining how variance is shown.
>
> As pointed out by another reviewer, the explanation of the concept of neuron's centre is already pretty terse and fast.  While for some readers just the math is enough, we feel that a graphical representation of the concept of the centre is the most straight forward way to present the meaning of \hat{w}-\hat{c}, which is critical to understanding how Insens works.
>
> In our opinion, a theorem is just a way of presenting a logical argument by making a statement followed by a proof.  Theorems are often used to prove trivial and intuitive results for the sake of completeness and unambiguity.  In our case, theorem is a clear way of presenting the building block of Insens.  Yes, the proof is elementary, but the theorem (even if obvious), is fundamental to the Insens method.  Indeed, once seen, our method for "collapsing" a deep network into a single linear projection might not seem that surprising.  The point is not that we merely show that it's possible, but what we do with it.  This seemingly straight forward observation leads us to decompose a computation in a neural network into active and inactive subnetworks.  We demonstrate how the inactive network, implicitly ignored by other interpretability methods, not only provides information for interpretability, but also can provide information about what happens inside the network.  Compared to DeepTaylor (which, as Figure 7 show is perhaps in the end not all that informative about what the model is doing), our method is mathematically simple and very intuitive - surely that's a good thing.  If the reviewer can provide a reference where switched linear projection has been presented, we'd be happy to reference it.  However, in the absence of prior publication using switched linear projections (or similar) for interpretability, we believe that, on the whole, we present something quite novel and not at all trivial.
>
> We'll be happy to make other minor fixes suggested by the reviewer.

---

> > ### Author Response · Authors · 2019-11-13
> > **With respect to variance in Figure #7...**
> >
> > We just realised what the reviewer meant by 20 repeated experiments of 100 samples for the correlation results.  It was not to get 20 times more samples to evaluate visualisation correlations between the trained, random and random label networks, but rather 20 evaulations on independently trained, randomly initialised and random label trained networks and then average that.   It could happen that a network with random weights happens to be sensitive to the same patterns that trained one...so it's better to do the experiment many times on the same architectures, but different instances of trained, random and randomly labelled networks.  We can definitely do that and report the results.

---

> > > ### Comment · AnonReviewer7 · 2019-11-13
> > > **Yes**
> > >
> > > Right, exactly that was I meant.

---

> > ### Comment · AnonReviewer7 · 2019-11-13
> > **Rebuttal**
> >
> > Thank you for the detailed response.
> >
> > Regarding the "outline" captured by DeepTaylor and Fig 5: For MNIST, Insens also captures the outline or the boundary. The reason I pointed out about Fig 5 being not very useful  is that it is hard for to see what is the intrepretation that is being done through this figure. You mention "Insens can show what each layer is doing in the neural network", but what is it showing is not clear to me. It is extracting out some plots from each layer, yes, which may be the other methods may not be able to, but could you say how what is being extracted is useful, even with context of capturing texture.
> >
> > Regarding the theorem 1 --- I should have been clearer, apologies. I think its a bit pedantic - about what is classified as a theorem and it can be a bit subjective. May be call it an "observation" or "proposition" rather than a "theorem" ?

---

> > > ### Author Response · Authors · 2019-11-14
> > > **Interpretability**
> > >
> > > We agree that it is not obvious what interpretation of visualisations in Figure 5 for, say, CIFAR10 images is, other than it doesn't look like the penultimate layer pays attention to shape.  However, visualisations from other layers from Figure 6 and sanity checks from Figure 7 provide a bit of context that is missing from other methods.  It is clear that the network gradually filters out information from the input and it doesn't look like this filtering is proceeding in an intuitive way - with the background being removed and the outline+texture of the object of interest remaining.  Instead, a peculiar combination of information from the object and its surroundings emerges as something that has high impact on the computation of the penultimate layer.  Still, the fact that Insens can extract and identify these gradual stages of raw input to abstraction, in our view, makes the inactive state analysis a powerful tool.
> > >
> > > We have no problem with changing the word "theorem" to "proposition" in the next update of the paper.

---

> > > > ### Comment · AnonReviewer7 · 2019-11-15
> > > > **Fig 5**
> > > >
> > > > My concern is with the value that Fig 5 adds, which seems there is none. There doesnt seem to be anything that suggests from Fig 5 that it helps with interpretation in anyway. Am I misunderstanding ? Your response seems to be "We dont know what Fig 5 is doing, but look at Fig 6,7" which seems to be tangential to what I am asking and trying to understand --- What is the value that Fig 5 adds in this paper ?

---

> > > > > ### Author Response · Authors · 2019-11-15
> > > > > **Fig 5**
> > > > >
> > > > > We would say that Figure 5 tells us what patterns in the image play an important role in the prediction.  It's just happens not to be an outline of the shape for the smallNORB and CIFAR10 examples.  The earlier figure with MNIST examples shows demonstrates that the method is capable of showing sensitivity to shape and outline when that's all there is in the input and that's what network uses to make its decision.  In Figure 5 we provide evaluation of Insens on harder datasets other than MNIST.  It establishes number of things - nice visualisations might not be telling as the "truth" of what the model is actually doing, as sanity checks on DeepTaylor demonstrate, and the patterns that the network uses from not clean dataset are...well, not very clean.  Still, we feel Insens visualisation provide somewhat better visualisations that other gradient-based methods while still passing the sanity checks.  At worst, they show what is important for the inactive state, which does affect the decision and is not picked up by other methods.  Finally, sanity checks in Figure 7 relate to visualisations shown in Figure 5 and Figure 6 - without Figure 5 this information has no context.  The purpose of Insens is to go after the patterns that "matter" to the network without making assumptions of what these patterns should be.

---

### Author Response · Authors · 2019-11-15
**Revision #3**

We would like to thank once more to all the reviewers for reading our paper and valuable suggestions.

The changes going into the third revision are:
- Numerous minor fixes suggested by the reviewers
- Reduction of the input examples in Figure 2 to two in order to free up space
- Sanity checks in Figure 7 done over 20 different instances of the 2CONV network
- Additional comment in the Conclusion about Insens and active neurons
- Expansion of Appendix C include multiple objects experiment with adversarial examples

---

### Decision · Program_Chairs · 2019-12-19

**Decision:**

Reject

**Comment:**

 This paper proposes a method to capture patterns of the so called “off” neurons using a newly proposed metric. The idea is interesting and worth pursuing. However, the paper needs another round of modification to improve both writing and experiments.